# Deep transfer learning for reducing health care disparities arising from biomedical data inequality

Yan Gao[1,2] & Yan Cui [1,2,3✉]

As artificial intelligence (AI) is increasingly applied to biomedical research and clinical decisions, developing unbiased AI models that work equally well for all ethnic groups is of crucial importance to health disparity prevention and reduction. However, the biomedical data inequality between different ethnic groups is set to generate new health care disparities through data-driven, algorithm-based biomedical research and clinical decisions. Using an extensive set of machine learning experiments on cancer omics data, we find that current prevalent schemes of multiethnic machine learning are prone to generating significant model performance disparities between ethnic groups. We show that these performance disparities are caused by data inequality and data distribution discrepancies between ethnic groups. We also find that transfer learning can improve machine learning model performance for data-disadvantaged ethnic groups, and thus provides an effective approach to reduce health care disparities arising from data inequality among ethnic groups.

[1] Department of Genetics, Genomics and Informatics, University of Tennessee Health Science Center, Memphis, TN 38163, USA. [2] Center for Integrative and Translational Genomics, University of Tennessee Health Science Center, Memphis, TN 38163, USA. [3] Center for Cancer Research, University of Tennessee Health Science Center, Memphis, TN 38163, USA. ✉email: ycui2@uthsc.edu

Artificial intelligence (AI) is fundamentally transforming biomedical research and health care systems are increasingly reliant on AI-based predictive analytics to make better diagnosis, prognosis, and therapeutic decisions[1–3]. Since data are the most important resources for developing high-quality AI models, data inequality among ethnic groups is becoming a global health problem in the AI era. Recent statistics showed that samples from cancer genomics research projects, including the TCGA[4], TARGET[5], OncoArray[6], and 416 cancer-related genome-wide association studies, were collected primarily from Caucasians (91.1%), distantly followed by Asians (5.6%), African Americans (1.7%), Hispanics (0.5%), and other populations (0.5%)[7]. Most clinical genetics and genomics data have been collected from individuals of European ancestry and ethnic diversity of studied cohorts has largely remained the same or even declined in recent years[8,9]. As a result, non-Caucasians, which constitute about 84% of the world's population, have a long-term cumulative data disadvantage. Inadequate training data may lead to nonoptimal AI models with low prediction accuracy and robustness, which may have profound negative impacts on health care for the data-disadvantaged ethnic groups[9,10]. Thus, data inequality between ethnic groups is set to generate new health care disparities.

The current prevalent scheme of machine learning with multiethnic data is the mixture learning scheme in which data for all ethnic groups are mixed and used indistinctly in model training and testing (Fig. 1). Under this scheme, it is unclear whether the machine learning model works well for all ethnic groups involved. An alternative approach is the independent learning scheme in which data from different ethnic groups are used separately to train independent models for each ethnic group (Fig. 1). This learning scheme also tends to produce models with low prediction accuracy for data-disadvantaged minority groups due to inadequate training data.

Here, we show that the mixture learning scheme tends to produce models with relatively low prediction accuracy for data-disadvantaged minority groups, due to data distribution mismatches between ethnic groups. Therefore, the mixture learning scheme often leads to unintentional and even unnoticed model performance gaps between ethnic groups. We find that the transfer learning[11–13] scheme (Fig. 1), in many cases, can provide machine learning models with improved performance for data-disadvantaged ethnic groups. Our results from machine learning experiments on synthetic data indicate that data inequality and data distribution discrepancy between different ethnic groups are the key factors underlying the model performance disparities. We anticipate that this work will provide a starting point for an unbiased multiethnic machine learning paradigm that implements regular tests of the performance of machine learning models on all ethnic groups to identify model performance disparities between ethnic groups, and that uses transfer learning or other techniques to reduce performance disparities. Such a paradigm is essential for reducing health care disparities arising from the long-standing biomedical data inequality among ethnic groups.

## Results

**Clinical omics data inequalities among ethnic groups.** Interrelated multi-omics factors including genetic polymorphisms, somatic mutations, epigenetic modifications, and alterations in expression of RNAs and proteins collectively contribute to cancer pathogenesis and progression. Clinical omics data from large cancer cohorts provide an unprecedented opportunity to elucidate the complex molecular basis of cancers[14–16] and to develop machine learning-based predictive analytics for precision oncology[17–22]. However, data inequality among ethnic groups continues to be conspicuous in recent large-scale genomics-focused biomedical research programs[7,23,24]. The TCGA cohort consists of 80.5% European Americans (EAs), 9.2% African Americans (AAs), 6.1% East Asian Americans (EAAs), 3.6% Native Americans (NAs), and 0.7% others, based on genetic ancestry analysis[25,26]. The TARGET[5] and MMRF CoMMpass[27] cohorts have similar ethnic compositions[28], which are typical for current clinical omics datasets[7]. The data inequality among ethnic groups is ubiquitous across almost all cancer types in the TCGA and MMRF CoMMpass cohorts (see Supplementary Fig. 1); therefore, its negative impacts would be broad and not limited to the cancer types or subtypes for which ethnic disparities have already been reported.

**Disparities in machine learning model performance.** We assembled machine learning tasks using the cancer omics data and clinical outcome endpoints[29] from the TCGA data of two ethnic groups: AA and EA groups, assigned by genetic ancestry analysis[25,26]. A total of 1600 machine learning tasks were assembled using combinations of four factors: (1) 40 types of cancers and pan-cancers[15]; (2) two types of omics features:

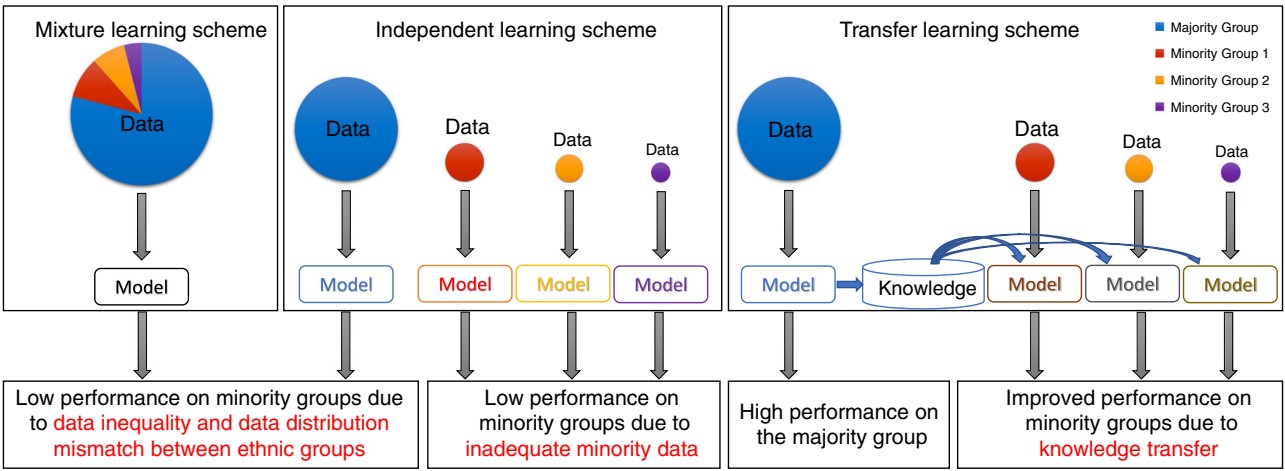

**Fig. 1 Multiethnic machine learning schemes.** In the mixture learning scheme, a model is trained and tested on the data for all ethnic groups. In the independent learning scheme, a model is trained and tested for each ethnic group using its own data. In the transfer learning scheme, a model is trained on the majority group data, then the knowledge learned is transferred to assist the development of a model for each minority group.

**Table 1 The machine learning experiments.**

| Multiethnic machine learning scheme | Experiment | Training data ethnic composition | Testing data ethnic composition | AUROC[a] | |
|---|---|---|---|---|---|
| | | | | **Median** | **Mean** |
| Mixture learning | Mixture 0 | AA + EA | AA + EA | 0.71 | 0.72 |
| | Mixture 1 | | EA | 0.71 | 0.73 |
| | Mixture 2 | | AA | 0.68 | 0.67 |
| Independent learning | Independent 1 | EA | EA | 0.70 | 0.71 |
| | Independent 2 | AA | AA | 0.59 | 0.58 |
| Transfer learning | Transfer learning | EA (source domain) AA (target domain) | AA | 0.70 | 0.69 |

[a]Median and mean AUROC (area under ROC curve) for each machine learning experiments on the 224 tasks.

mRNA and protein expression; (3) four clinical outcome endpoints: overall survival (OS), disease-specific survival (DSS), progression-free interval (PFI), and disease-free interval (DFI)[29]; and (4) five thresholds for the event time associated with the clinical outcome endpoints (Supplementary Fig. 2). For each learning task, each patient is assigned to a positive (or a negative) prognosis category based on whether the patient's event time for the clinical outcome endpoint of the learning task is no less than (or less than) a certain threshold.

Since the AA patients consist of less than 10% of the TCGA cohort, there were only very small numbers of AA cases in many learning tasks. We filtered out the learning tasks having too few cases to permit reliable machine learning experiments. We then performed machine learning experiments on the remaining 447 learning tasks that had at least five AA cases and five EA cases in each of the positive and negative prognosis categories. For each machine learning task, we trained a deep neural network (DNN) model for classification between the two prognosis categories using the mixture learning scheme. The mixture learning models achieved reasonably good baseline performance ($AUROC > 0.65$) for 224 learning tasks. A total of 21 types of cancers and pan-cancers and all four clinical outcome endpoints were represented in these learning tasks. The proportion of AA patients ranged from 0.06 to 0.25 in these learning tasks with a median of 0.12 (Supplementary Fig. 3a). For each of the 224 learning tasks (Supplementary Data 1), we performed six machine learning experiments (Table 1) to compare the performance of the three multiethnic machine learning schemes on the AA and EA groups (Fig. 2).

In the machine learning experiments, we observed that the mixture learning scheme was prone to produce biased models with a lower prediction performance for the data-disadvantaged AA group. The model performance differences between the EA and AA groups were statistically significant with a $p$ value of $6.72 \times 10^{-11}$ (Fig. 2, Mixture 1 & 2). The average EA–AA model performance gap over the 224 learning tasks was 0.06 (AUROC, Table 1). Without testing the model performance of the machine learning models on each ethnic group separately, the performance differences would be concealed by the overall good performance for the entire multiethnic cohort (Fig. 2, Mixture 0). The independent learning scheme produced even larger EA–AA performance differences with a $p$ value of $1.29 \times 10^{-26}$ and the average performance gap was 0.13 (Table 1, Fig. 2, Independent 1 & 2).

**Transfer learning for improving machine learning model performance for data-disadvantaged ethnic groups.** We compared machine learning schemes on performance for the data-disadvantaged AA group and found that transfer learning produced models with significantly better performance for the AA

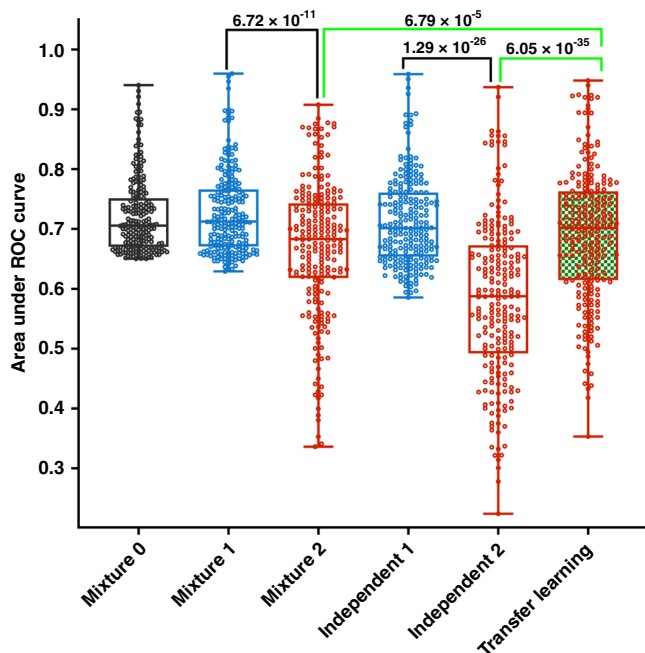

**Fig. 2 Performance index values for the multiethnic machine learning experiments.** Each box plot shows the AUROC (area under ROC curve) values for the 224 learning tasks for a machine learning experiment listed in Table 1. Each circle represents the mean AUROC of 20 independent runs with different random partitions of training and testing data. The gray color represents performance for the whole cohort, blue represents performance for the EA group, and red represents performance for the AA group. Box-plot elements are: center line, median; box limits, 25 and 75 percentiles; whiskers, the minimum and maximum values. The $p$ values were calculated using one-sided Wilcoxon signed-rank test.

group compared to the models from mixture learning ($p = 6.79 \times 10^{-5}$) and independent learning ($p = 6.0.5 \times 10^{-35}$) (Fig. 2). The machine learning experiment results for four learning tasks with different cancer types and clinical outcome endpoints are shown in Fig. 3 (more results in Supplementary Fig. 4). We used threefold cross-validation and performed 20 independent runs for each experiment using different random partitions of training and testing data to assess machine learning model performance. The median AUROC of the six experiments are denoted as $A_{Mixture0}$, $A_{Mixture1}$, $A_{Mixture2}$, $A_{Independent1}$, $A_{Independent2}$, and $A_{Transfer}$. The results of these experiments showed a consistent pattern:

(1) Both mixture learning and independent learning schemes produced models with relatively high and stable performance

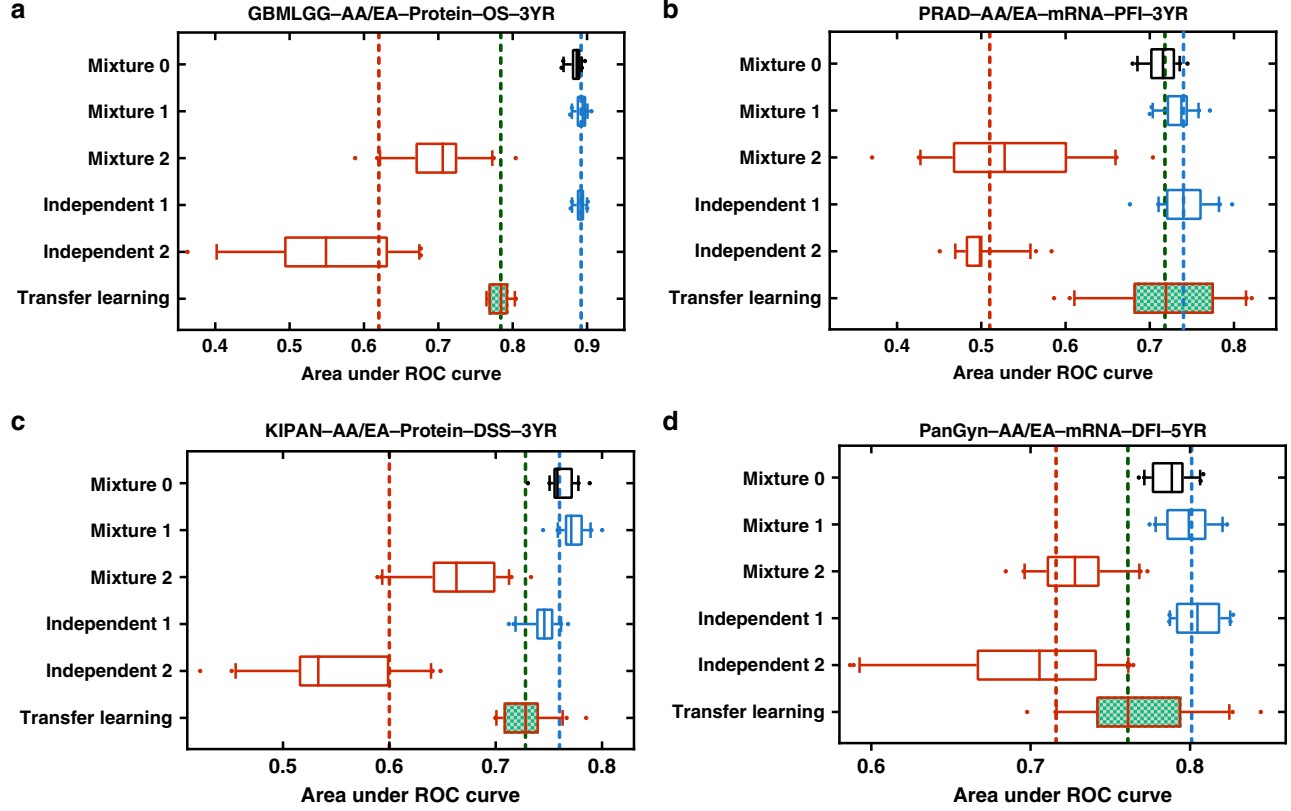

**Fig. 3 Comparison of multiethnic machine learning schemes.** The machine learning tasks are: **a** GBMLGG-AA/EA-Protein-OS-3YR, **b** PRAD-AA/EA-mRNA-PFI-3YR, **c** KIPAN-AA/EA-Protein-DSS-3YR, **d** PanGyn-AA/EA-mRNA-DFI-5YR. In each panel, the box plots show AUROC values for the six experiments (20 independent runs for each experiment). The red, blue, and green vertical dash lines represent $\overline{AUROC}_{AA}$, $\overline{AUROC}_{EA}$, and $A_{Transfer}$ respectively. Box-plot elements are: center line, median; box limits, 25 and 75 percentiles; whiskers, 10–90 percentiles; points, outliers. Abbreviations for cancer types are explained in Supplementary Data 1.

for the EA group but low and unstable performance for the data-disadvantaged ethnic group (AA). We defined the performance disparity gap as $G = \overline{AUROC}_{EA} - \overline{AUROC}_{AA}$, where $\overline{AUROC}_{EA} = (A_{Mixture1} + A_{Independent1})/2$, and $\overline{AUROC}_{AA} = (A_{Mixture2} + A_{Independent2})/2$. $G$ is represented by the distance between the blue and red dash lines in Fig. 3 and Supplementary Fig. 4.

(2) The transfer learning scheme produced models with improved performance for the data-disadvantaged AA group, and thus reduced the model performance gap. The reduced model performance disparity gap is $\tilde{G} = \overline{AUROC}_{EA} - A_{Transfer}$, which is represented by the distance between the blue and green dash lines in Fig. 3 and Supplementary Fig. 4.

Among the 224 learning tasks, 142 had a performance gap $G > 0.05$ and 88.7% (125/142) of these performance gaps were reduced by transfer learning.

We also performed the machine learning experiments on two additional learning tasks that involved either another ethnic group or non-TCGA data: (1) Stomach Adenocarcinoma (STAD)-EAA/EA-PFI-2YR assembled using the TCGA STAD data of EAA and EA patients; and (2) MM-AA/EA-mRNA-OS-3YR assembled using the MMRF CoMMpass[27] data of AA and EA patients (Supplementary Data 1). For both learning tasks, machine learning experiments showed the same pattern of performance as described above (Supplementary Fig. 4a, b).

**Key factors underlying ethnic disparities in machine learning model performance.** A machine learning task $\mathcal{T} =$

$\{\mathcal{X}, \mathcal{Y}, f : \mathcal{X} \to \mathcal{Y}\}$ consists of a feature space $\mathcal{X}$, a label space $\mathcal{Y}$, and a predictive function $f$ learned from feature-label pairs. From a probabilistic perspective, $f$ can be written as[13] $P(Y|X)$, where $X \in \mathcal{X}$, and $Y \in \mathcal{Y}$. It is generally assumed that each feature-label pair is drawn from a single distribution[30] $P(X, Y)$. However, this assumption needs to be tested for multiethnic omics data. Given $P(X, Y) = P(Y|X)P(X)$, both marginal distribution $P(X)$ and the conditional distribution $P(Y|X)$ may contribute to the data distribution discrepancy among ethnic groups. We used $t$-test to identify differentially expressed mRNAs or proteins between the AA and EA groups. The median percentage of differentially expressed mRNA or protein features in the 224 learning tasks was 10%, and 70% of the learning tasks had at least 5% differentially expressed mRNA or protein features (Supplementary Fig. 3b). We used logistic regression to model the conditional distribution $f = P(Y|X)$, and calculated the Pearson correlation coefficient between the logistic regression parameters for the AA and EA groups. The Pearson correlation coefficients ranged from $-0.14$ to $0.26$ in the learning tasks, with a median of $0.04$ (Supplementary Fig. 3c). These results indicate that various degrees of marginal and conditional distribution discrepancies between the AA and EA groups exist in most of the 224 learning tasks.

We hypothesized that the data inequality represented by cohort ethnic composition and data distribution discrepancy between ethnic groups are the key factors underlying the ethnic disparity in machine learning model performance and that both factors can be addressed by transfer learning. To test this hypothesis, we performed the six machine learning experiments (Table 1) on synthetic data generated using a mathematical model whose parameters represent these hypothetical key factors (Methods).

Synthetic Data 1 was generated using parameters estimated from the data for the learning task PanGyn-AA/EA-mRNA-DFI-5YR (Fig. 3d), which simulated data inequality and distribution discrepancy between the ethnic groups in the real data (Table 2). For this synthetic dataset, the six machine learning experiments showed a performance pattern (Fig. 4a) similar to that of the real data (Fig. 3), which was characterized by performance gaps from the mixture and independent learning schemes and by transfer learning reduction of the performance gaps. Synthetic Data 2 has no distribution difference between the two ethnic groups (Table 2). For this dataset, there is no performance gap from the mixture learning scheme, however, the performance gap from the independent learning scheme remains (Fig. 4b). Synthetic Data 3 has equal numbers of cases from the two ethnic groups

(no data inequality) but has a distribution discrepancy between the two ethnic groups. Synthetic Data 4 has equal numbers of cases from the two ethnic groups (no data inequality) and does not have a distribution difference between the two ethnic groups. For these two datasets, there is no significant performance gap from any learning scheme (Fig. 4c, d). These results confirm that the performance gap from the mixture learning scheme is caused by both data inequality and data distribution discrepancy between ethnic groups while the performance gap from the independent learning scheme is caused by inadequate data for the disadvantaged ethnic group, and transfer learning may reduce these performance gaps (Fig. 1).

## Discussion

In this work, we show that the current prevalent scheme for machine learning with multiethnic data, the mixture learning scheme, and its main alternative, the independent learning scheme, tend to generate machine learning models with relatively low performance for data-disadvantaged ethnic groups due to inadequate training data and data distribution discrepancies among ethnic groups. We also find that transfer learning can provide improved machine learning models for data-disadvantaged ethnic groups by leveraging knowledge learned from other groups having more abundant data. These results indicate that transfer learning can provide an effective approach to reduce health care disparities arising from data inequality among ethnic groups. Our simulation experiments show that the machine learning performance disparity gaps would be eliminated completely if there was no data inequality regardless of data distribution discrepancies (Table 2, Fig. 4c, d). Algorithm-based

**Table 2 Multiethnic machine learning experiments on synthetic data.**

| Synthetic data | Data inequality | Distribution discrepancy | Machine learning model performance gap | |
| --- | --- | --- | --- | --- |
| | | | Mixture learning | Independent learning |
| 1 | Yes | Yes | Yes[a] | Yes[a] |
| 2 | Yes | No | No | Yes[a] |
| 3 | No | Yes | No | No |
| 4 | No | No | No | No |

[a]Performance gap > 0.05 (AUROC).

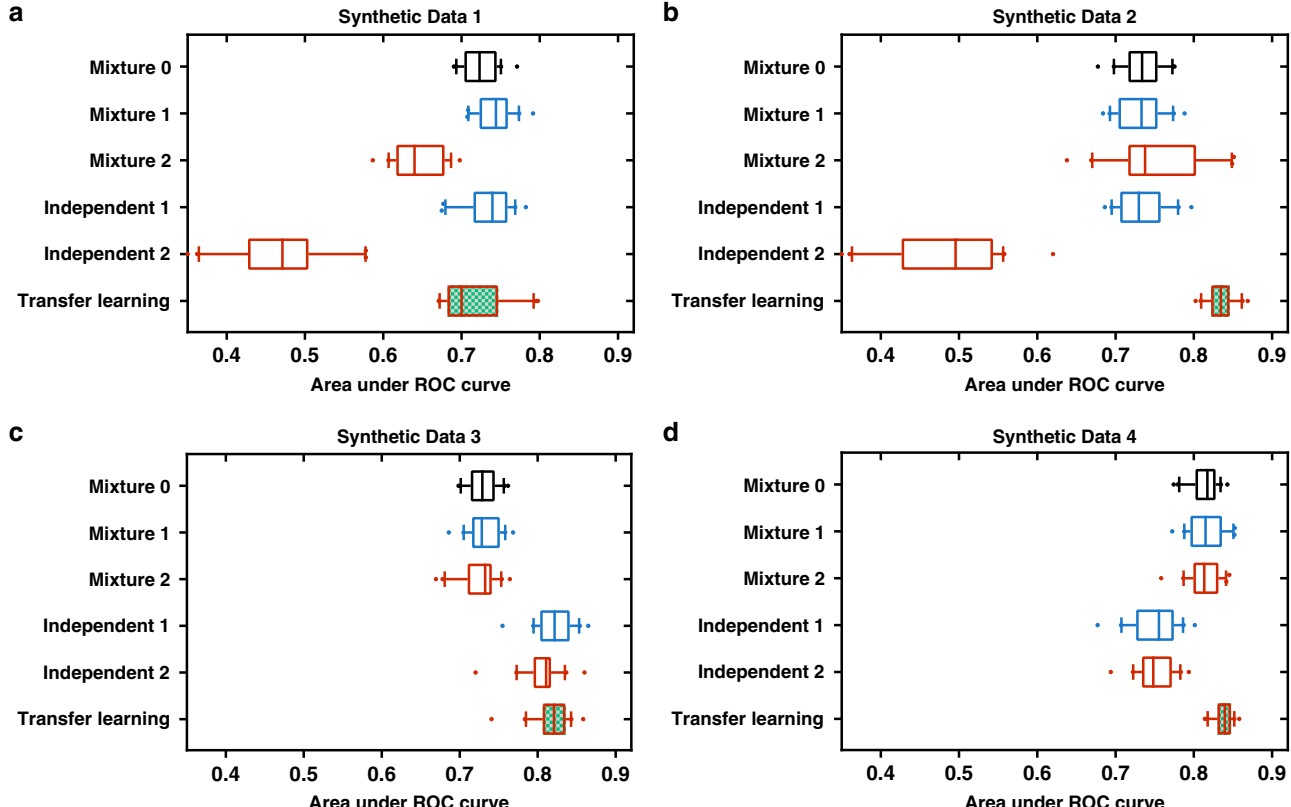

**Fig. 4 Comparison of multiethnic machine learning schemes on synthetic data. a** Synthetic Data 1, **b** Synthetic Data 2, **c** Synthetic Data 3, **d** Synthetic Data 4. We used threefold cross-validation and performed 20 independent runs for each experiment with different random partitions of training and testing data to assess machine learning model performance. In each panel, box plots show the AUROC values for the six experiments (20 independent runs for each experiment). Box-plot elements are: center line, median; box limits, 25 and 75 percentiles; whiskers, 10–90 percentiles; points, outliers.

methods may mitigate health care disparities arising from long-standing data inequality among ethnic groups; however, the ultimate solution to this challenge would be to increase the number of minority participants in clinical studies.

Many factors, including ethnic composition of the cohort, omics data type, cancer type, and clinical outcome endpoint, may potentially affect the performance of multiethnic machine learning schemes. At this point, it is not clear how these factors affect the performance of transfer learning and other learning schemes. One possible direction for future research is to discover how the performance pattern of multiethnic learning schemes changes as a function of these factors.

## Methods

**Data source and data preprocessing**. The TCGA and MMRF CoMMpass data used in this work were downloaded from the Genome Data Commons (GDC, https://gdc.cancer.gov). The ethnic groups of TCGA patients were determined based on the genetic ancestry data downloaded from The Cancer Genetic Ancestry Atlas[25] (TCGAA, http://52.25.87.215/TCGAA). The ethnic groups of MMRF CoMMpass patients were based on the self-reported information in the clinical data file downloaded from the GDC Data Portal (https://portal.gdc.cancer.gov).

For the TCGA data, we used all the 189 protein expression features, and the 17176 mRNA features without missing values. We further removed samples with more than 20% missing values. We also filtered out samples missing genetic ancestry or clinical endpoint data. The data matrix was standardized such that each feature has a zero mean and unit standard deviation. The ANOVA $F$ value for each mRNA was calculated for the training samples to select 200 mRNAs as the input features for machine learning. The feature mask, ANOVA $F$ value, and $p$ values were calculated using the SelectKBest function (with the f_classif score function and $k = 200$) of the python sklearn package[31]. For the MMRF CoMMpass data, we selected 600 mRNA features with the highest mean absolute deviation as the input features for machine learning.

**Deep neural network modeling**. We used the Lasagne (https://lasagne.readthedocs.io/en/latest/) and Theano python packages (http://deeplearning.net/software/theano/) to train the DNN. We used a pyramid architecture[32] with 6 layers: an input layer with 200 nodes for mRNA features or 189 nodes for protein features, 4 hidden layers including a fully connected layer with 128 nodes followed by a dropout layer[33], a fully connected layer with 64 nodes followed by a dropout layer, and a logistic regression output layer. To fit a DNN model, we used the stochastic gradient descent method with a learning rate of 0.01 ($lr = 0.01$) to find the weights that minimized a loss function consisting of a cross-entropy and two regularization terms: $l(W) = -\sum_{i=1}^{m} (y_i\log(\hat{y}_i) + (1 - y_i)\log(1 - \hat{y}_i)) + \lambda_1|W| + \lambda_2\|W\|_2$, where $y_i$ is the observed label of patient $i$, $\hat{y}_i$ is the predicted label for patient $i$, and $W$ represents the weights in the DNN. Traditional activation functions such as the sigmoid and hyperbolic tangent functions have a gradient vanish problem in training a deep-learning model, which may lead to gradient decreasing quickly and training error propagating to forward layers. Here, we use the ReLU function $f(x) = \max(0, x)$, which is widely used in deep learning to avoid the gradient vanish problem. For each dropout layer, we set the dropout probability $p = 0.5$ to randomly omit half of the weights during the training to reduce the collinearity between feature detectors. To speed up the computation, we split the data into multiple mini-batches during training. We used a batch size of 20 (batch_size = 20) for two basic learning schemes (mixture learning and independent learning for the EA group) as there were relatively large numbers of cases available for training. For the independent learning for the AA group, we set the batch size to 4 because the number of cases available for training was limited. We set the maximum number of iterations at 100 (max_iter = 100) and applied the Nesterov momemtum[34] method (with momentum = 0.9 for each DNN model) to avoid premature stopping. We set the learning rate decay factor at 0.03 (lr_decay = 0.03) for the learning task BRCA-AA/EA-Protein-OS-4YR to avoid non-convergence during training. For all other tasks, we set lr_decay = 0. The two regularization terms $\lambda_1$ and $\lambda_2$ were set at 0.001.

**Transfer learning**. For transfer learning[11–13,35–37], we set the EA group as the source domain and the AA or EAA group as the target domain. We applied three transfer learning methods to each learning task and selected the best AUROC as the performance index for the transfer learning scheme. The three transfer learning methods include two fine-tuning algorithms and a domain adaptation algorithm:

(1) Fine-tuning algorithm 1
Recent studies have shown that fine-turning of DNN often leads to better performance and generalization in transfer learning[38]. We first pretrained a DNN model using source domain data: $M \sim f(Y_{Source}|X_{Source})$, which has the same architecture as described in the previous section. The training parameters were set as lr = 0.01, batch_size = 20, p = 0.5, max_iter = 100, and momentum = 0.9. After the initial training, the DNN model was then

fine-tuned using backpropagation on the target domain data: $M' = fine\_tuning (M|Y_{Target}, X_{Target})$, where $M'$ was the final model. In the fine tuning, the learning rate was set at 0.002 and the batch size was set at 10 as the model had been partially fitted and the target dataset was small.

(2) Fine-tuning algorithm 2
In the second fine-tuning algorithm, the source domain data were used as unlabeled data to pretrain a stacked denoising autoencoder[37,39,40]. The stacked denoising autoencoder has 5 layers: the input layer, a coding layer with 128 nodes, a bottleneck layer with 64 nodes, a decoding layer with 128 nodes, and an output layer that has the same number of nodes with the input layer to reconstruct the input data. We used the source and target domain data to train the stacked autoencoder with the parameters: learning rate = 0.01, corruption level = 0.3, batch size = 32, and maximum iteration = 500. After pretraining the autoencoder, we removed the decoder and added a dropout layer (with p = 0.5) after each hidden layer, and then added a fine-tune (logistic regression) layer. The final DNN model had the same architecture as described in the previous section and was fine-tuned on target domain data with training parameters lr = 0.002, batch_size = 10 and max_iter = 100.

(3) Domain adaptation
Domain adaptation is a class of transfer learning methods that improve machine learning performance on the target domain by adjusting the distribution discrepancy across domains[41,42]. We adopted the Contrastive Classification Semantic Alignment (CCSA) method[43] for domain adaptation. The CCSA method is particularly suitable for our transfer learning tasks because: (1) this method can significantly improve target domain prediction accuracy by using very few labeled target samples for training; (2) this method includes semantic alignment in training and therefore can handle the domain discrepancy in both marginal and conditional distributions. To use the CCSA method which calculates the pairwise Euclidean distance between samples in the embedding space, we applied a $L2$ norm transformation to the features of each patient such that for patient $i$, $\sum_{j=1}^{n} x_{ij}^2 = 1$, where $n$ is the number of features. The CCSA minimizes the loss function $L_{CCSA}(f) = (1 - \gamma)L_C(h \circ g) + \gamma(L_{SA}(h) + L_S(g))$, where $f = h \circ g$ is the target function, $g$ is an embedding function that maps the input $X$ to an embedding space $Z$, and $h$ is a function to predict the output labels from $Z$, $L_C(f)$ denotes the classification loss (binary cross-entropy) of function $f$, $L_{SA}(h)$ refers to the semantic alignment loss of function $h$, $L_S(g)$ is the separation loss of function $g$, $\gamma$ is the weight used to balance the classification loss versus the contrastive semantic alignment loss $L_{SA}(h) + L_S(g)$, $L_{SA}(h) = \frac{1}{n} \sum_{y_i^s = y_j^t} \frac{1}{2} \|g(x_i^s), g(x_j^t)\|^2$ and $L_S(g) = \frac{1}{n} \sum_{y_i^s \neq y_j^t} \frac{1}{2}\max(0, m - \|g(x_i^s), g(x_j^t)\|)^2$, $\|.\|$ is the Euclidean distance, while $m$ is the margin that specifies the separability of the two domain features in the embedding space[43]. During the training, we set the parameters $m = 0.3$, momentum = 0.9, batch_size = 20, learning_rate = 0.01, and max_iter = 100. We used one hidden layer with 100 nodes for semantic alignment and added a dropout layer ($p = 0.5$) after the hidden layer for classification.

**Differential expression analysis**. For each learning task, we performed a permutation-based $t$-test on the input features to select the proteins or mRNAs that were differentially expressed between the AA and EA groups. The mRNAs and proteins with a feature-wise $p$ value < 0.05 were selected as differentially expressed features between the two ethnic groups.

**Logistic regression**. For each learning task, we fit two multivariate logistic regression models: $Y^{AA} = 1/(1 + e^{-\beta^{AA} \cdot X^{AA}})$, $Y^{EA} = 1/(1 + e^{-\beta^{EA} \cdot X^{EA}})$, for the AA group and the EA group, respectively, to calculate the regression parameters for each ethnic group.

**Stratified cross-validation and training/testing data for machine learning experiments**. For each learning task, we applied a threefold stratified cross-validation[44]. For mixture learning, samples were stratified by the clinical outcome and genetic ancestry in the process of threefold data splitting. Samples of each fold had the same distribution over clinical outcome classes (positive and negative) and ethnic groups (EA and AA). Both AA and EA samples in the training set were used to train a deep-learning model and the performance of Mixture 0 was measured using the whole testing set, the performance of Mixture 1 was measured on the EA samples in the testing set, and the performance of Mixture 2 was measured on the AA samples in the testing set. For Independent learning, EA (Independent 1) and AA (Independent 2) samples were separated and then stratified by the clinical outcome in the threefold data splitting. The cross-validation was performed for the two ethnic groups separately. For transfer learning, EA and AA samples were separated and AA samples were stratified by the clinical outcome (same as Independent 2), and we used all the EA (source domain) samples for initial model training and then used AA training samples for fine-tuning or domain adaptation, and finally, the performance was evaluated on AA testing samples. The ethnic

**Table 3 Parameters used to generate the synthetic data.**

| Synthetic data | Data inequality parameters | | Distribution discrepancy parameters | | | | |
|---|---|---|---|---|---|---|---|
| | $n_1$ | $n_2$ | $n_{de}$ | $n_{-1,-1}$ | $n_{-1,1}$ | $n_{1,-1}$ | $n_{1,1}$ |
| 1 | 218 | 43 | 20 | 64 | 37 | 37 | 62 |
| 2 | 218 | 43 | 0 | 100 | 0 | 0 | 100 |
| 3 | 260 | 260 | 20 | 64 | 37 | 37 | 62 |
| 4 | 260 | 260 | 0 | 100 | 0 | 0 | 100 |

compositions for the training and testing data of the six types of machine learning experiments are shown in Table 1.

**Machine learning performance evaluation**. The main utility of performance metric in this work is to compare the relative performance of multiethnic machine learning schemes. We used the area under ROC curve[45] (AUROC) to evaluate performance of machine learning models. Another widely used machine learning performance metric is the area under precision–recall curve[46] (AUPR). It has been mathematically proven that the performance ranks of two models remain same in the ROC space and the PR space[47]. However, linear interpolation in the precision–recall space is problematic, which may lead to inaccurate calculation of AUPR for datasets of small sample sizes[47]. AUROC is a more robust metric for evaluating machine learning performance on the minority ethnic groups that have less cases.

**Synthetic data generator**. We developed a mathematical model to generate synthetic data for the multiethnic machine learning experiments. The simulated cohort consists of two ethnic groups. The degree of data inequality is controlled by the parameters: $n_1$ and $n_2$, which represent the numbers of individuals in the two ethnic groups. We used the ssizeRNA package[48] to generate the feature matrix $x_{ij}$. The number of differentially expressed features ($n_{de}$) is the parameter controlling marginal distribution ($P(X)$) discrepancy between the two ethnic groups. For individual $i$ in ethnic group $k$, the label $y_i^k$ was generated using the logistic regression function: $y_i^k = \begin{cases} 1 & if\ z_i^k > c^k \\ -1 & otherwise \end{cases}$, where $z_i^k = \frac{1}{1+e^{-\sum_{j=1}^{n} \beta_j^k x_{ij}}}$, $x_{ij}$ is the $j$th feature of individual $i$, and $\beta_j^k \epsilon \{-1, 1\}$ represents the effect of feature $j$ on the label of ethnic group $k$, and $c^k$ is the threshold for assigning a sample to the positive or negative category. A pair of $\beta_j^1$ and $\beta_j^2$ have four possible combinations representing the difference and similarity of the effect of feature $j$ on the clinical outcome for the patients in the two ethnic groups. The number of features associated with each of the four combinations is denoted as $n_{-1,-1}$, $n_{-1,1}$, $n_{1,-1}$, and $n_{1,1}$ respectively. These parameters control the conditional distribution ($P(Y|X)$) discrepancy between the two ethnic groups. Using this model, we can generate synthetic datasets with or without data inequality and/or distribution discrepancy between two ethnic groups by setting the parameter values. These parameters can also be estimated from a real dataset. For example, we generated Synthetic Data 1 using the parameters estimated from the data for the learning task PanGyn-AA/EA-mRNA-DFI-5YR. We set $n_1$ and $n_2$ to be equal to the number of EA and AA patients in the real data, respectively. We estimated the parameters $n_{de}$ using permutation-based $t$-tests (feature-wise $p$ value < 0.05). The total number of features for the learning task PanGyn-AA/EA-mRNA-DFI-5YR was 200. We used multivariate logistic regression to calculate the regression parameters $\boldsymbol{\beta}^{AA}$ and $\boldsymbol{\beta}^{EA}$. We let $\beta_j^1 = \begin{cases} 1 & if\ \beta_j^{EA} > \text{median}(\boldsymbol{\beta}^{EA}) \\ -1 & otherwise \end{cases}$ and $\beta_j^2 = \begin{cases} 1 & if\ \beta_j^{AA} > \text{median}(\boldsymbol{\beta}^{AA}) \\ -1 & otherwise \end{cases}$, and then calculated $n_{-1,-1}$, $n_{-1,1}$, $n_{1,-1}$, and $n_{1,1}$. The parameters used to generate Synthetic Data 1–4 are shown in Table 3.

**Reporting summary**. Further information on research design is available in the Nature Research Reporting Summary linked to this article.

## Data availability
The TCGA and MMRF CoMMpass datasets are publicly available at the Genome Data Commons (https://gdc.cancer.gov/about-data/publications/pancanatlas and https://gdc.cancer.gov/about-gdc/contributed-genomic-data-cancer-research/foundation-medicine/multiple-myeloma-research-foundation-mmrf). The processed datasets that were used as the input files for the machine learning experiments are available at https://doi.org/10.6084/m9.figshare.12811574.

## Code availability
Source code is available at https://github.com/ai4pm/TL4HDR.

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

## Acknowledgements

This research was supported by the Center for Integrative and Translational Genomics at University of Tennessee Health Science Center.

## Author contributions

Y.C. conceived and designed the study. Y.G. performed the data processing and the machine learning experiments. Y.G. wrote the computer code and the documentation. All authors developed the algorithms, interpreted the results and wrote the paper. All authors reviewed and approved the final paper.

## Competing interests

The authors declare no competing interests.
