## [Peer Review File · Nature Communications]

Reviewers' Comments:

Reviewer #1:

Remarks to the Author:

The authors present an interesting manuscript on an important topic that examines the impact of machine learning on racial disparities in model performance due to data inequality across racial groups. The authors also report findings that suggest “transfer learning” can improve machine learning model performance for groups poorly represented in data sets. Overall, these findings are informative particularly for efforts to reduce racial disparities in model (prognostic and/or treatment) generation from genomic data. Still, the findings remain limited by the problem that inspired the work itself (limited number of African American cases in TCGA), and it is difficult to conclude from these data that one model of machine learning outperforms another in minority groups. The authors should consider the following comments in preparing their revision:

-Page 3-4: Please make note of how racial groups are defined in TCGA (methods make note that the TCGAA was used). Also, it is worth noting that a significant proportion of patients in TCGA have unknown race (>10%)—this is a major limitation of TCGA and any race-based studies from the data set, and it is a strength of this manuscript that the authors used the TCGAA which accounts for a significant proportion of missing data. Also, please reference the study that created this database: <https://pubmed.ncbi.nlm.nih.gov/30300578/>

-The major limitation of this study is the small number of African American cases used for the machine learning experiments. This limitation is magnified by the number of machine learning experiments examined as well as the number of comparisons made across learning models (mixture, independent, and transfer). The authors should discuss their results within the context of these limitations, and should discuss methods that they used to reduce bias from multiple testing in the setting of small N. For example, for the test comparing Mixture 2 to the Transfer Learning model, the p-value for significance is 0.044—please make note if a correction for multiple testing was used.

-The authors should consider also making note that ultimately the best method to reduce the potential for racial disparities arising from lack of data in minority groups, is ultimately to increase the number of minority participants in these studies (a short sentence in the conclusion sentence may be the best place).

Reviewer #2:

Remarks to the Author:

This manuscript describes an elegant investigation of the impact of racial/ethnic cancer genomics data inequalities on artificial intelligence (specifically machine learning based in deep neural networks) predictive modelling and demonstrates the preferability of deep transfer learning to mixture or independent learning for modelling minority group outcomes. The authors use extant data sources (The Cancer Genome Atlas, Multiple Myeloma Research Foundation CoMMpass study) to illustrate the considerable biases in play with standard approaches and simulated data to show how both data inequality and data distribution discrepancies contribute to those biases. This work suggests that deep transfer learning may be an important way to ameliorate modelling biases with the potential to contribute to health care disparities.

The findings were, in general, well described and discussed. The figures were particularly clearly presented and paired nicely with table summaries. The manuscript could be further strengthened with some additional revisions.

1. Title (p. 1): This needs to be revised to better reflect the manuscript content. First, while the experiments described demonstrate that deep transfer learning can significantly improve AI

modelling biases, it is a stretch to claim that improved model performance will “reduce health disparities”. In addition, per the methods description on p. 4, the learning tasks described in the manuscript involve mRNA and protein expression, not genomic, data.

2. Abstract (p. 1): Again, the invocation of “genomic data inequality” here seems inapt given the data used in the study; broader reference to biomedical, rather than genomic, data would be more appropriate. In addition, “transfer learning” is not defined or clearly contrasted with the other major machine learning approaches (mixture, independent) explored in the manuscript.

3. Page 2, end of first paragraph, also end of second paragraph: Please replace the sentences that refer to “health disparities” with “health care disparities” instead. (and elsewhere in the manuscript, as appropriate)

4. Page 3, Genomic data inequalities among racial groups section: Again, it would seem better to word this section header more broadly to reflect the broader focus on clinical omics data.

5. Page 4, line 4 of second paragraph): It seems worthwhile to point out that the 447 learning tasks used for analysis were only a minority (28%) of the 1600 learning tasks assembled. What impact, if any, might that have had on the overall observations or conclusions?

6. Page 10, Methods, Data sources section: It was quite surprising to read that the racial/ethnic identities of patients whose data were used in the study were based in estimates of genetic ancestry, rather than self-reported or assigned racial/ethnic identity, at least for the TCGA data. This should be made clear earlier in the manuscript.

Reviewer #3:

Remarks to the Author:

Gao and Cui present a deep neural network-based transfer learning approach aimed at improving binary classifier performance for racial groups that are underrepresented in genomic studies. In this framework, the source domain is comprised of data from the genetic ancestry group with the most samples and the target domain is comprised of data from an ancestry group that is underrepresented in the genomic study. The authors compare the transfer learning approach to training on mixtures of ancestry groups and training on each group separately in a series of binary classification tasks using real (mRNA and protein expression) and synthetic data. They demonstrate that training on a mixture of groups masks lower performance in groups with fewer samples. In the synthetic data experiment, the authors demonstrate that differences in sample size (termed “data inequality” by the authors) and discrepancy between the distribution of features contribute to loss of performance when trained on a mixture of groups or when trained on the group with a smaller sample size independently.

The concept of the study is appealing, is in an important area of study, and I appreciate the inclusion of experiments to assess what factors affect machine learning performance. I have reservations about the manuscript, many of which arise from lack of clarity or framing of the problem but I find they negatively impact my ability to assess the manuscript in its current form.

Major

1. A major weakness of the paper is that the experimental design and methodology, at points, are insufficiently described or unclear. For example, it is not clear how the p-values presented in Figure 2 were calculated, how the training and test datasets described in Table 1 were constructed, how many evaluations used data from the MMRF CoMMpass study, and how the balance of classes were handled during 3-fold CV. Without sufficient detail, it can be challenging to fully assess the study.

2. The authors refer to European American and African American as two racial groups from the TCGA study. However, the authors state in the methods “The racial groups of TCGA patients were determined based on the genetic ancestry data downloaded from The Cancer Genetic Ancestry Atlas (TCGAA, <http://52.25.87.215/TCGAA>).” These two groups appear to refer to genetic ancestry

derived from EIGENSTRAT, rather than self-reported race which is what I expected as a reader based on the chosen terminology. The references included in the introduction seem to be centered on self-reported race and self-reported ethnicity (Guerrero et al. Scientific Reports. 2018.) or genetic ancestry (Genetics for all. Nature Genetics. 2019.; Martin AR et al. Nature Genetics. 2019.) The authors should avoid conflating these terms and be specific about what information was used to construct groups in this study. Clarity on what is being evaluated is particularly important in the context of any discussion of health disparities (see Mersha and Abeba. Human Genomics. 2015. DOI: 10.1186/s40246-014-0023-x).

3. The authors use an approach based on deep neural network modeling, which likely requires considerable computational resources and expertise to apply. Is it possible to achieve the objective of the study or similar performance using a “simpler” approach? For example, using a principal component analysis-based approach to remove variation associated with genetic ancestry comes to mind (concept similar to Price et al. Nature Genetics. 2006. DOI: 10.1038/ng1847).

4. A single metric for evaluating model performance is presented in throughout the paper (AUROC). I don't believe that the authors must present an exhaustive set of metrics to support their conclusions, but some discussion of the suitability or rationale for the use of AUROC in this context would be welcome. Specifically, is this metric going to be appropriate for all of the tasks presented, where there may be severe class imbalance in some of the tasks (Saito and Rehmsmeier. PLoS One. 2015. DOI: 10.1371/journal.pone.0118432)? As a reader, I have some ideas about why AUROC might have been chosen—total accuracy is inappropriate because of class imbalance (which is unlikely to be the same between the groups being tested); it seems reasonable that we care about positive and negative classes equally. However, it would be helpful for the authors to state their rationale and any limitations.

5. From Figure 2, it looks like there are a number of instances where the performance of the transfer learning approach is poor (AUROC < 0.65). Do these tasks where performance is poor have some kind of commonality (e.g., always use protein expression) and if so, do the authors have recommendations for when transfer learning is not suitable for a task?

Minor

1. A URL for the GitHub repository referenced in the data and code availability section should be included.

2. Authors should consider using a repository specifically for research artifacts such as figshare or Zotero, which have the benefits of DOI and versioning, rather than a Dropbox link to distribute data required for running the pipeline. I would recommend including a specific GitHub release of the source code in either of these repositories as well.

3. The labeling on Supplementary Figure 3 could be improved by adding y-axis labels and titles.

4. There appear to be two items labeled as Supplementary Table 1 – one in the Excel spreadsheet that contains all other Supplementary Tables and one that contains information about the 224 ML tasks included.

5. The differential expression analysis section references miRNAs; I believe this should be mRNA based on other parts of the manuscript.

Response to reviewers' comments

We thank the reviewers for their helpful comments. We have revised the manuscript in response to the comments and suggestions.

Reviewer #1:

-Page 3-4: Please make note of how racial groups are defined in TCGA (methods make note that the TCGAA was used). Also, it is worth noting that a significant proportion of patients in TCGA have unknown race (>10%)—this is a major limitation of TCGA and any race-based studies from the data set, and it is a strength of this manuscript that the authors used the TCGAA which accounts for a significant proportion of missing data. Also, please reference the study that created this database: <https://pubmed.ncbi.nlm.nih.gov/30300578/>

Response: We agree that genetic ancestry often provides more accurate and complete data compared with self-reported racial information. We have revised the description for racial composition of TCGA on Page 3 based on the genetic ancestry analysis results of Yuan et al. and have cited the paper, which is an important reference.

-The major limitation of this study is the small number of African American cases used for the machine learning experiments. This limitation is magnified by the number of machine learning experiments examined as well as the number of comparisons made across learning models (mixture, independent, and transfer). The authors should discuss their results within the context of these limitations, and should discuss methods that they used to reduce bias from multiple testing in the setting of small N. For example, for the test comparing Mixture 2 to the Transfer Learning model, the p-value for significance is 0.044—please make note if a correction for multiple testing was used.

Response: Most cancer clinical omics datasets, including the TCGA and MMRF CoMMpass datasets contain only relatively small numbers of cases from the racial minority groups. This data inequality is exactly the challenge we want to address in this work. Our simulation results showed that the performance gaps from the mixture and independent learning schemes would vanish if there was no data inequality (Fig.4C&D). The performance comparison between different learning methods over multiple datasets (in our case 224 datasets) is not a multiple testing problem because only one statistical test was performed for each of the four pairwise comparisons (Fig.2). In this revised manuscript, we used the Wilcoxon Signed-Rank test, which is more appropriate for algorithm performance comparison over multiple datasets/tasks as recommended in the highly cited paper of J. Demšar (Statistical comparisons of classifiers over multiple data sets, Journal of Machine Learning Research 7:1-30, 2006). For the comparison between Transfer Learning and Mixture 2, the p-value calculated from the Wilcoxon Signed-Rank test is 6.79×10^{-5} , which indicates that the performance improvement from the transfer learning scheme is statistically significant. The previously used (unpaired) t-test underestimated the statistical significance.

-The authors should consider also making note that ultimately the best method to reduce the potential for racial disparities arising from lack of data in minority groups, is ultimately to increase the number of minority participants in these studies (a short sentence in the conclusion sentence may be the best place).

Response: Great point. We have added a sentence in the Conclusion section: “Algorithm-based methods may mitigate health care disparities arising from long-standing data inequality among racial/ethnic groups; however, the ultimate solution to this challenge would be to increase the number of minority participants in clinical studies” (Page 10).

Reviewer #2:

1. *Title (p. 1): This needs to be revised to better reflect the manuscript content. First, while the experiments described demonstrate that deep transfer learning can significantly improve AI modelling biases, it is a stretch to claim that improved model performance will “reduce health disparities”. In addition, per the methods description on p. 4, the learning tasks described in the manuscript involve mRNA and protein expression, not genomic, data.*

Response: We have changed the title to “deep transfer learning for reducing health care disparities arising from *biomedical* data inequality”. We think there is a clear link between “improved machine learning model performance for the data-disadvantaged racial/ethnic minorities” and “reducing health care disparities arising from data inequality” because health care systems are increasingly reliant on AI-based predictive analytics to make better diagnosis, prognosis and therapeutic decisions¹⁻³.

2. *Abstract (p. 1): Again, the invocation of “genomic data inequality” here seems inapt given the data used in the study; broader reference to biomedical, rather than genomic, data would be more appropriate. In addition, “transfer learning” is not defined or clearly contrasted with the other major machine learning approaches (mixture, independent) explored in the manuscript.*

Response: We agree that “biomedical data inequality” is a more appropriate term and have made the revisions suggested by the reviewer. The three multiethnic machine learning schemes are defined and contrasted with each other in Fig 1 and its caption and are also described in the second paragraph on Page 2. The details of the algorithms are described in the Method section. We have added a section to clarify on the cross-validation process and the training/testing data used for these multiethnic machine learning schemes (Page 13-14)

3. *Page 2, end of first paragraph, also end of second paragraph: Please replace the sentences that refer to “health disparities” with “health care disparities” instead. (and elsewhere in the manuscript, as appropriate)*

Response: We agree that “biomedical data inequality” is a more accurate term and have made the revisions suggested by the reviewer.

4. *Page 3, Genomic data inequalities among racial groups section: Again, it would seem better to word this section header more broadly to reflect the broader focus on clinical omics data.*

Response: We have revised the section title using the broader term “clinical omics”.

5. *Page 4, line 4 of second paragraph): It seems worthwhile to point out that the 447 learning tasks used for analysis were only a minority (28%) of the 1600 learning tasks assembled. What impact, if any, might that have had on the overall observations or conclusions?*

Response: We filtered out the learning tasks that have too few cases for reliable machine learning experiments. The filtering criteria are described on Page 4 (the 2nd paragraph). This is a standard unbiased filtering process that should not affect our conclusions. As we mentioned in the manuscript, the 224 learning tasks in the final analysis broadly covered a total of 21 types of cancers and pan-cancers and all four clinical outcome endpoints from the TCGA data.

6. *Page 10, Methods, Data sources section: It was quite surprising to read that the racial/ethnic identities of patients whose data were used in the study were based in estimates of genetic ancestry, rather than self-reported or assigned racial/ethnic identity, at least for the TCGA data. This should be made clear earlier in the manuscript.*

Response: We have revised the description for the racial composition of TCGA on Page 3&4 and mentioned that the racial information was derived from genetic ancestry analysis.

Reviewer #3:

Major

1. A major weakness of the paper is that the experimental design and methodology, at points, are insufficiently described or unclear. For example, it is not clear how the p-values presented in Figure 2 were calculated, how the training and test datasets described in Table 1 were constructed, how many evaluations used data from the MMRF CoMMpass study, and how the balance of classes were handled during 3-fold CV.

Response: We have clarified these issues and added methodology details in the revised manuscript. The p-values in the revised Figure 2 were calculated using the one-sided Wilcoxon signed-rank test, which is widely used for classifier performance comparison over multiple datasets/tasks. As mentioned in the manuscript, in addition to the 224 tasks, we also performed the machine learning experiments on two additional learning tasks that involved either a racial minority group other than AA or non-TCGA data. One of these two tasks was assembled from the MMRF CoMMpass data, which is a much smaller dataset (compared to TCGA) containing only one feature type (mRNA) and one clinical endpoint (OS). We used a stratified cross-validation process to reduce the bias introduced by regular cross-validation methods which randomly splits the dataset into k folds without considering class distributions. We have added a section to describe the stratified cross-validation and training/testing data for machine learning experiments (Page 13-14).

2. The authors refer to European American and African American as two racial groups from the TCGA study. However, the authors state in the methods “The racial groups of TCGA patients were determined based on the genetic ancestry data downloaded from The Cancer Genetic Ancestry Atlas (TCGAA, <http://52.25.87.215/TCGAA>).” These two groups appear to refer to genetic ancestry derived from EIGENSTRAT, rather than self-reported race which is what I expected as a reader based on the chosen terminology. The references included in the introduction seem to be centered on self-reported race and self-reported ethnicity (Guerrero et al. *Scientific Reports*. 2018.) or genetic ancestry (*Genetics for all*. *Nature Genetics*. 2019.; Martin AR et al. *Nature Genetics*. 2019.) The authors should avoid conflating these terms and be specific about what information was used to construct groups in this study. Clarity on what is being evaluated is particularly important in the context of any discussion of health disparities (see Mersha and Abeba. *Human Genomics*. 2015. DOI: 10.1186/s40246-014-0023-x).

Response: We agree that self-reported race/ethnicity data has its limitations, but genotype data are not always available to estimate genetic ancestry. The papers cited in the Introduction section showed a consistent picture of biomedical data inequality among racial/ethnic groups regardless of the analyses using either genetic ancestry or self-reported racial information. The genetic ancestry of the MMRF CoMMpass patients could not be determined because genotype data were not available. All the 224 learning tasks and one of the two additional leaning tasks were assembled from TCGA data for which genetic ancestry was used. We have clarified that self-reported race information was used for the MMRF CoMMpass patients (Page 10). The terms (European American and African American) are consistent with those used in the TCGA genetic ancestry analysis paper (Yuan et al. *Cancer Cell*. 2018 doi:10.1016/j.ccell.2018.08.019) and the TCGAA Database (<http://52.25.87.215/TCGAA/summary.php>), which is the source of the genetic ancestry data used in this work.

3. The authors use an approach based on deep neural network modeling, which likely requires considerable computational resources and expertise to apply. Is it possible to achieve the objective of the study or similar performance using a “simpler” approach? For example, using a

principal component analysis-based approach to remove variation associated with genetic ancestry comes to mind (concept similar to Price et al. Nature Genetics. 2006. DOI: 10.1038/ng1847).

Response: Our goal is to improve machine learning model performance on data-disadvantaged racial minority groups. We do not know if there is any “simpler” method that could achieve the same objective. EIGENSTRAT (Price et al., 2006) is a PCA-based method for population stratification using genotype data. Genetic ancestries of the TCGA patients used in this work were calculated using EIGENSTRAT and STRUCTURE (Pritchard et al., 2000) by Yuan et al (Cancer Cell. 2018). At this point, we do not see any direct application of such methods to reducing the negative impact of data inequality on machine learning.

4. A single metric for evaluating model performance is presented in throughout the paper (AUROC). I don't believe that the authors must present an exhaustive set of metrics to support their conclusions, but some discussion of the suitability or rationale for the use of AUROC in this context would be welcome. Specifically, is this metric going to be appropriate for all of the tasks presented, where there may be severe class imbalance in some of the tasks (Saito and Rehmsmeier. PLoS One. 2015. DOI: 10.1371/journal.pone.0118432)? As a reader, I have some ideas about why AUROC might have been chosen—total accuracy is inappropriate because of class imbalance (which is unlikely to be the same between the groups being tested); it seems reasonable that we care about positive and negative classes equally. However, it would be helpful for the authors to state their rationale and any limitations.

Response: We agree that accuracy is obviously not an appropriate performance metric because of the class imbalance issue. We also agree that one of the reasons to use ROC is that the positive and negative classes are both important for our learning tasks. The major alternative to the ROC curve is the Precision-Recall (PR) curve. It has been suggested that the PR plot is more informative than the ROC plot for the evaluation of binary classifiers on imbalanced datasets, especially when the number of negatives outweighs the number of positives significantly (Saito and Rehmsmeier. PLoS One. 2015). However, most (145 out of the 224) of our machine learning tasks have more positive samples than negative samples. More importantly, the major utility of the performance metric in this work is to compare the relative performance of the machine learning models from different learning schemes. It has been mathematically proven that “for a fixed number of positive and negative examples, one curve dominates a second curve in ROC space *if and only if* the first dominates the second in Precision-Recall space” (Theorem 3.2. in Davis and Goadrich, 2006)⁵, which means the performance ranks of two models should be the same regardless of using ROC or PR. Finally, an intrinsic property of the PR curve makes it unsuitable for evaluating performance on the minority groups that have small sample sizes. As shown in the Davis and Goadrich paper⁵: “In Precision-Recall space, interpolation is more complicated. As the level of Recall varies, the Precision does not necessarily change linearly due to the fact that FP replaces FN in the denominator of the Precision metric. In these cases, linear interpolation is a mistake that yields an overly-optimistic estimate of performance.” This property of the PR curve will lead to an inaccurate calculation of the area under PR curve if the dataset has a small sample size, which means having only a small number of points in the PR curve. ROC curves do not have this problem.

5. From Figure 2, it looks like there are a number of instances where the performance of the transfer learning approach is poor (AUROC < 0.65). Do these tasks where performance is poor have some kind of commonality (e.g., always use protein expression) and if so, do the authors have recommendations for when transfer learning is not suitable for a task?

Response: Many factors, including racial composition, feature type, cancer type and clinical outcome endpoint, could potentially affect the performance of the multiethnic machine learning

schemes. At this point, we do not have any empirical rule that can be used to predict the performance of transfer learning for a task. In practice, one can always compare the performance of transfer learning on a data-disadvantaged minority group with the performance of other learning schemes on the same racial group (e.g. Mixture 2 and Independent 2) and select the best method.

Minor

1. A URL for the GitHub repository referenced in the data and code availability section should be included.

Response: We have deposited all the data and code in GitHub and added the URL for the GitHub repository to the manuscript.

2. Authors should consider using a repository specifically for research artifacts such as figshare or Zotero, which have the benefits of DOI and versioning, rather than a Dropbox link to distribute data required for running the pipeline. I would recommend including a specific GitHub release of the source code in either of these repositories as well.

Response: We have deposited the source data files to GitHub and Figshare.

3. The labeling on Supplementary Figure 3 could be improved by adding y-axis labels and titles.

Response: We have added descriptive y-axis labels to Supplementary Figure 3.

4. There appear to be two items labeled as Supplementary Table 1 – one in the Excel spreadsheet that contains all other Supplementary Tables and one that contains information about the 224 ML tasks included.

Response: Supplementary Table 1 has three sheets that are labeled as “The 224 learning tasks”, “Two additional learning tasks” and “Cancer Type Abbreviations” respectively.

5. The differential expression analysis section references miRNAs; I believe this should be mRNA based on other parts of the manuscript.

Response: Yes, it should be mRNA. We have made the correction.

Reference

1. Topol, E.J. High-performance medicine: the convergence of human and artificial intelligence. *Nature Medicine* **25**, 44-56 (2019).
2. Azuaje, F. Artificial intelligence for precision oncology: beyond patient stratification. *NPJ precision oncology* **3**, 6 (2019).
3. Rajkomar, A., Dean, J. & Kohane, I. Machine Learning in Medicine. *New England Journal of Medicine* **380**, 1347-1358 (2019).
4. Breiman, L., Friedman, J., Stone, C.J. & Olshen, R.A. *Classification and regression trees*, (CRC press, 1984).
5. Davis, J. & Goadrich, M. The relationship between Precision-Recall and ROC curves. in *Proceedings of the 23rd international conference on Machine learning* 233-240 (2006).

Reviewers' Comments:

Reviewer #1:

Remarks to the Author:

The authors have sufficiently addressed my reviewer comments.

Reviewer #2:

Remarks to the Author:

No additional comments

Reviewer #3:

Remarks to the Author:

In this revision, the authors have addressed initial comments by adding a new methods section, including the statistical test used in the Figure 2 legend, and making clarifications about the use of genetic ancestry assignments for the TCGA data prior to the methods section. Below are comments on the revision.

- Some comments on the initial submission (e.g., use of differences between AUROC as a metric, limitations of transfer learning for some tasks where performance remains poor), although addressed in the response to reviewers, have not resulted in substantive changes to the text of the manuscript that would benefit readers.
- The GitHub and figshare links appear to 404 in the PDF version of the manuscript. This needs to be addressed in order for the work to be reproducible.

We thank the reviewer for the helpful comments. We have revised the manuscript in response to the comments and suggestions.

Reviewer #3:

- Some comments on the initial submission (e.g., use of differences between AUROC as a metric, limitations of transfer learning for some tasks where performance remains poor), although addressed in the response to reviewers, have not resulted in substantive changes to the text of the manuscript that would benefit readers.

Response: We have added a section on machine learning performance evaluation to discuss the performance metric (Page 14). As we mentioned in the previous response to reviewers' comments, many factors could potentially affect the performance of multiethnic machine learning schemes and we do not know any empirical rule for how these factors may affect the performance of transfer learning and other learning schemes at this point. However, this could be an interesting direction for future research. We have added a brief discussion on this issue (the last paragraph in the Discussion section on Page 10).

- The GitHub and figshare links appear to 404 in the PDF version of the manuscript. This needs to be addressed in order for the work to be reproducible.

Response: We have fixed the issue.